# TwinCons: Conservation score for uncovering deep sequence similarity and divergence

**Petar I. Penev** [1,2], **Claudia Alvarez-Carreño** [1,3], **Eric Smith** [1,4,5,6,7], **Anton S. Petrov** [1,3]*, **Loren Dean Williams** [1,2,3]*

**1** NASA Center for the Origin of Life, Georgia Institute of Technology, Atlanta, Georgia, United States of America, **2** School of Biological Sciences, Georgia Institute of Technology, Atlanta, Georgia, United States of America, **3** School of Chemistry and Biochemistry, Georgia Institute of Technology, Atlanta, Georgia, United States of America, **4** Earth-Life Science Institute, Tokyo Institute of Technology, Meguro-ku, Tokyo, Japan, **5** Santa Fe Institute, Santa Fe, New Mexico, United States of America, **6** Department of Physics, The University of Wisconsin-Madison, Madison, Wisconsin, United States of America, **7** Ronin Institute, Montclair, New Jersey, United States of America

* anton.petrov@biology.gatech.edu (ASP); loren.williams@chemistry.gatech.edu (LDW)

**Data Availability Statement:** The TwinCons source code is available at https://github.com/LDWLab/TwinCons The source code also includes scripts used for dataset processing and classifier training and evaluation. TwinCons is also available as a

## Abstract

We have developed the program TwinCons, to detect noisy signals of deep ancestry of proteins or nucleic acids. As input, the program uses a composite alignment containing predefined groups, and mathematically determines a 'cost' of transforming one group to the other at each position of the alignment. The output distinguishes conserved, variable and signature positions. A signature is conserved within groups but differs between groups. The method automatically detects continuous characteristic stretches (segments) within alignments. TwinCons provides a convenient representation of conserved, variable and signature positions as a single score, enabling the structural mapping and visualization of these characteristics. Structure is more conserved than sequence. TwinCons highlights alternative sequences of conserved structures. Using TwinCons, we detected highly similar segments between proteins from the translation and transcription systems. TwinCons detects conserved residues within regions of high functional importance for the ribosomal RNA (rRNA) and demonstrates that signatures are not confined to specific regions but are distributed across the rRNA structure. The ability to evaluate both nucleic acid and protein alignments allows TwinCons to be used in combined sequence and structural analysis of signatures and conservation in rRNA and in ribosomal proteins (rProteins). TwinCons detects a strong sequence conservation signal between bacterial and archaeal rProteins related by circular permutation. This conserved sequence is structurally colocalized with conserved rRNA, indicated by TwinCons scores of rRNA alignments of bacterial and archaeal groups. This combined analysis revealed deep co-evolution of rRNA and rProtein buried within the deepest branching points in the tree of life.

python package in the Python Package Index https://pypi.org/project/TwinCons/. The script used for calculating rRNA TwinCons distributions is available at https://github.com/LDWLab/TWC_distribution. The github projects were archived at https://zenodo.org/record/4654505. Alignment datasets Training datasets, query composite alignments, and the caspase-metacaspase alignment are available at: https://apollo2.chemistry.gatech.edu/TwinConsDatasets/.

**Funding:** This work was funded by the National Aeronautics and Space Administration grant 80NSSC18K1139 awarded to LDW and ASP. CA-C was supported by the NASA Postdoctoral Program fellowship award. The funders had no role in study design, data collection and analysis, decision to publish, or preparation of the manuscript.

**Competing interests:** The authors have declared that no competing interests exist.

## Author summary

All species on Earth can be thought of as leaves on the Tree of Life, which are connected by branches representing their ancestral relationships. Biopolymers are evolutionary markers within species, that contain records of evolutionary history. Excavation of molecular evolutionary histories involves collecting sequences from extant species and organizing them into multiple sequence alignments. For the purpose of comparison, the sequences within an alignment can be partitioned into two groups, resulting in a composite alignment. We have developed the program TwinCons, to detect noisy signals of deep ancestry. TwinCons distinguishes conserved, variable and signature positions between the groups of the composite alignment. A signature is a position conserved within each group but differing between groups. TwinCons can further be used to detect uninterrupted ranges of positions (segments) preserved within the composite alignment. TwinCons results can be mapped onto structures of molecules. TwinCons scores can be applied to either proteins or ribonucleic acids (RNA). Using TwinCons we detected highly similar segments across ancient and essential protein components of living cells (translation and transcription) and pinpointed the deepest signatures between bacterial and archaeal RNAs within the ribosome.

This is a *PLOS Computational Biology* Methods paper.

## Introduction

Advances in sequencing and metagenomics [1] allow biologists to observe alterations in cancer genomes [2], identify genetic novelty [3], characterize microbial communities [4], fold protein sequences [5], and unravel ancestral relationships [6–8]. Sequence data are commonly analyzed and interpreted in the form of multiple sequence alignments in which rows are linear sequences and columns are analyzed to quantitate conservation, which can indicate homology and common ancestry [9].

### Composite alignments

We wish to probe ancestral relationships within the translation and transcription systems, which contain some of the oldest proteins and RNAs in the biological world. Can we look back far beyond the Last Universal Common Ancestor (LUCA) [10] and establish whether segments of ribosomal proteins (rProteins) share ancestry with segments of RNA polymerase (RNAP) or translation factors, and determine how these segments interplay with nearly RNA? For that purpose, here we establish and apply a score of conservation within a composite alignment, which is a multiple sequence alignment containing two pre-defined sequence groups. For example, in a composite alignment of a single gene, one group could be bacterial sequences while the other could be archaeal sequences. Alternatively, a composite alignment might contain paralogous sequences, where two different but related genes form the groups. One group could be an initiation factor and the other group could be an elongation factor. In this scenario, the homology within each group is known or suspected, but homology between the groups is in question. The method developed here, called TwinCons, can be applied to either protein or nucleic acid sequences. Classical methods consider only conserved and non-

conserved alignment positions, while TwinCons differentiates between universal, signature, and non-conserved segments or positions.

## Phylogenetic signatures

Universal positions are conserved within and between groups. We define and detect signature positions as those that are conserved within groups but differ between groups. Signature positions have high mutational cost between groups as defined by substitution matrices [11–13]. Protein signature positions that exhibit radical differences between groups have been referred to as "functionally divergent" [14,15] or "constant-but-different" [16] or "subfamily specific" [17,18] or positions that maximize functional similarity within groups and functional variation between groups [19]. Such positions hint at differential mechanisms of function between groups. Signature positions that exhibit differences in group-specific evolutionary rates suggest residues that are critical for the function of one group but not the other [20–22].

## Search for ancient conservation

Classical methods for homology detection are based on alignments of a query sequence and a database of annotated sequences, using sequence similarity scores or sequence profiles. Some of the most widely used methods to detect homology employ local sequence similarity and Hidden Markov Models [23–31]. These methods, together with annotated motif descriptor databases are the basis of protein classification systems, and of function and structure prediction algorithms [32]. These methods have become standards in annotating novel sequences, and they infer ancestry of long conserved regions. Traditional methods rarely produce positive results from deep and noisy ancestry that is locked in short segments of proteins with different functions and structures. In these cases, researchers directly inspect conservation levels within alignments to discover specific functional sites and homologies within a group of sequences [33–37].

Alignments provide a variety of information such as relative rates of mutation. Substitution models are obtained by determining which substitutions are common or rare in alignments of homologous sequences [38–42]. Further sequestering alignments by different functions or structure classes allows construction of substitution models of specific structural elements [43] or among specialized types of proteins (e.g. membrane proteins [44], mitochondrial proteins [45]). Henikoff & Henikoff [46] used distantly related sequences to calculate BLOSUM matrices, which do not implement any evolutionary model, but are rather a direct observation on substitution frequencies within alignment sequence blocks.

The neutral theory of molecular evolution [47] states that structurally and/or functionally important sites within proteins experience stronger selection, with lower rates of change. Therefore, high conservation has been widely used as an indicator for functionally important sites within proteins [37,48–50]. Numerous studies have applied this theory by calculating evolutionary rates from alignments based on specific positions [51,52], specific structures [43], or global phylogenies [33]. Statistical methods have been developed to detect sequence divergencies between duplicated genes [15,53,54] and have been used to discover functionally important changes between proteins [55,56].

## The TwinCons score

In the current work we describe TwinCons–which quantifies sequence conservation and signature positions in protein or nucleic acid composite alignments. TwinCons highlights universally conserved, variable, and signature positions in alignments of enzymatic and non-

enzymatic proteins. TwinCons detects short segments that are common between proteins. Such short segments are hypothesized to have recorded the deepest relationships between proteins [57,58]. Using TwinCons 'full capabilities allowed us to discover correlations between highly similar rProtein segments and ribosomal RNA (rRNA) signatures in the most ancient ribosomal components. The TwinCons method incorporates a composite alignment, with groups for each polymer of interest. The method uses prior information on ancestry within each group. Instead of calculating statistics of similarity, we mathematically determine a 'cost' of transforming an alignment column of one group to the corresponding alignment column of the other group.

TwinCons offers advantages over previous methods. TwinCons identifies and discriminates between universally conserved sites (large positive scores), intra-group signature sites (large negative scores) and random sites (near zero scores). A positive TwinCons value corresponds to conservation within each group in addition to conservation between the groups. A negative TwinCons value represents conservation within groups and divergence between groups. Finally, a TwinCons value near zero represents variability within and between groups.

## Materials and methods

### TwinCons calculation methodology

**Summary.** TwinCons computes and compares, for each position of a composite alignment, two 20-dimensional vectors of amino acid frequencies, one for each group. For nucleic acid alignments the vectors are in 4-dimensions. TwinCons extends the formalism of a pairwise alignment to composite alignment between two groups (See Supplementary text in S1 Appendix for a detailed description). TwinCons computes the price of transformation of one vector onto the other, filtered by a substitution matrix. The nature of the conservation is defined by the substitution matrix, and can be based on empirical mutational frequencies, physico-chemical properties, etc. The scores computed over all columns within a composite alignment can be used to identify conserved regions, signature regions and random regions between the two groups. Thus, the proposed score provides multiple utilities, and is useful for comprehensive analysis of deep protein ancestry.

**Selection of groups within the alignment.** TwinCons is a metric that estimates the similarity between two sequence groups for each position within a composite alignment considering global substitution frequencies inferred from a substitution matrix (Table A in S1 Appendix). Similarity is determined by the values of off diagonal elements in the substitution matrix. For each alignment position two distributions of frequencies are calculated based on the two groups: Group I and Group II that are defined prior to TwinCons calculations. Groups can be manually defined in the input alignment or computed using the deepest branching point of a phylogenetic tree, built from the input alignment.

**Gap adjustment in the composite alignment.** The TwinCons score performs gap adjustment by prorating their frequency. Prorating can be done uniformly (for protein and nucleic acid alignments) or by using the background frequency of a specified substitution matrix (only for protein alignments, default option). With uniform prorating option, a single gap character counts as 0.05 of every amino acid residue (for a protein alignment) or 0.25 of every nucleotide (for nucleic acid alignments). With frequency prorating option, each gap contributes a skewed frequency distribution from all 20 amino acids, provided from the substitution matrix. The gap adjusted frequencies for each group are then computed for every column of the alignment and used for the TwinCons calculations.

Heavily gapped columns can be ignored by removing them from the calculation. The default gap percentage threshold (GT) for a column to be removed is calculated as:

$$GT = \min\left(\frac{G1}{G1 + G2}, \frac{G2}{G1 + G2}\right) - 0.05 \tag{1}$$

where G1 and G2 denote the number of sequences present in each group and min is a function that takes the smaller result. This ensures that regions present only in one group are removed and is most useful when mapping results on three-dimensional structures. Gapped regions in an alignment can be omitted with the option "-cg". The option "-gt" followed by percentage as a decimal number (e.g., 0.9 = 90% gaps) can be used to override the default percentage for removal of columns.

**Vector frequency calculations.**   We represent the distributions of the gap adjusted frequencies in each column of Groups I and II as n-dimensional vectors Xn and Yn. With "n" we denote the number of possible residues (n = 4 in nucleotide alignments and n = 20 in amino-acid alignments). Each value in the vector is the fractional occurrence for a given residue, therefore the sum of values within one vector always equals one.

**Frequency weighting.**   Each sequence within a group optionally can be weighted based on its distance from the other sequences using either a Voronoi algorithm [59] or distance based on branch length of a tree [60]. Thus, if an alignment contains many identical sequences, they will be given less importance.

**Substitution matrices in TwinCons.**   TwinCons supports a variety of substitution matrices in log-odds form and is designed to work both with nucleotide and protein alignments. For nucleotides we can select between a simple identity matrix, a transition/transversion matrix and the *blastn* matrix (Table A in S1 Appendix). For amino acids we support a variety of substitution matrices (Table A in S1 Appendix) [61]. Many of the available matrices are evolutionary informed by mutational rates from global alignments (like Blosum62) and their off-diagonal values confer information about signature substitutions. When provided with a structure file we use structurally informed substitution matrices that account for the partitioning of sequence into secondary and 3D structure information (Table A in S1 Appendix) [43]. Custom matrices in the PAML triangular format are also supported [62,63]. Throughout the manuscript we use structurally informed LG-derived substitution matrices when a structure is available for both groups within a composite alignment. When structures are not available, we use the LG matrix which does no structural partitioning.

**TwinCons calculations.**   To calculate the cost of transforming one frequency vector in another we take the transpose of one of the vectors and dot multiply both vectors by the scoring matrix. Thus, for each alignment position TwinCons (TWC) is calculated as:

$$TWC = Y_n * M * X_n^T \tag{2}$$

where $X_n^T$ is a column vector of frequency components for Group I, $Y_n$, is a raw vector of frequency components for Group II, and M is a transformation matrix (the cost of the transformation), which determines the mutation penalties between each pair of residues or nucleotides. For a complete mathematical justification of the TwinCons calculation see Supplementary text in S1 Appendix. The scoring matrices M are symmetric (equal to their transpose), therefore TwinCons does not discriminate between departure and arrival points. In a given composite alignment, TwinCons is computed for each position of the alignment.

**Baseline correction.**   The range of TwinCons values depends on the matrix used. To have an ability of comparing TwinCons scores computed using different matrices, we performed baseline correction. Each matrix is adjusted by adding a value to each of its elements such that

the TwinCons computed for a selected distribution is zero. The value is calculated either uniformly or, when available, using the background frequency of the substitution matrix (option "-bn"). The default behavior is to use background frequency. In both cases the baseline correction aims to produce a matrix which evaluates to 0 when dot multiplied with the baselining vectors. When using uniform baseline correction TwinCons is comparable to conservation scores, except that it accounts for multiple groups. Classical conservation scores show maximum entropy, no information and zero conservation for completely random positions. When using background frequency baseline correction TwinCons produces scores which are more realistic for biological sequences.

Baseline correction follows naturally from the interpretation of TwinCons as a log-likelihood score for paired sets of aligned residues, sampled from the leaves of a branching process generated by the chosen substitution matrix. The baseline derived from the substitution matrix is simply the expectation of TwinCons on a random sample, and thus the null hypothesis under that generating model for samples grouped in the same way as the data. The derivation of TwinCons as a statistic, including the baseline correction, and possible compositional adjustments [64,65] is provided in the Supplementary text in S1 Appendix.

No additional normalization is performed to the maximal and minimal intensities across different matrices. These values are a meaningful description of the evolutionary processes present in the original datasets used to build the matrices. The TwinCons score tries to maximally preserve this information.

**Example.** For example, an alignment position has the nucleotide distribution 'GAT-TACA' in one group and 'AAAAAAA' in the other group, this will produce a pair of 4-dimensional vectors (Eq 3).

$$GCAT : [0.14, 0.14, 0.44, 0.28] \; (GATTACA) \tag{3}$$

$$GCAT : [0, 0, 1, 0] \; (AAAAAAA)$$

These vectors can be represented as points in a 4-dimensional space; to reach one point from the other we must traverse the space between them. To represent the costs associated with traversing that distance we use an n-by-n scoring matrix. In our example we use a 4-by-4 matrix defined from the blastn algorithm and adjusted to produce score zero for uniform vectors. TwinCons for our example vectors evaluates to 1.71 (Eq 4).

$$|0.14 \; 0.14 \; 0.44 \; 0.28| * \begin{vmatrix} 6.75 & -2.25 & -2.25 & -2.25 \\ -2.25 & 6.75 & -2.25 & -2.25 \\ -2.25 & -2.25 & 6.75 & -2.25 \\ -2.25 & -2.25 & -2.25 & 6.75 \end{vmatrix} * \begin{vmatrix} 0 \\ 0 \\ 1 \\ 0 \end{vmatrix} = 1.71 \tag{4}$$

## Evaluating group similarity with TwinCons

To evaluate similarity between groups from a composite alignment we calculate segments based on TwinCons scores. A segment is defined as a range of positions, within which the cumulative TwinCons score is continuously increasing. To calculate a cumulative representation of the score we consecutively add the TwinCons score for each position. Next, the cumulative score is smoothed using the Savitzky-Golay filter [66]. Segments are detected by identifying local minima and maxima of the smoothed cumulative score. That way we generate positive uninterrupted segments (Fig 1C). Each alignment generates a distribution of these

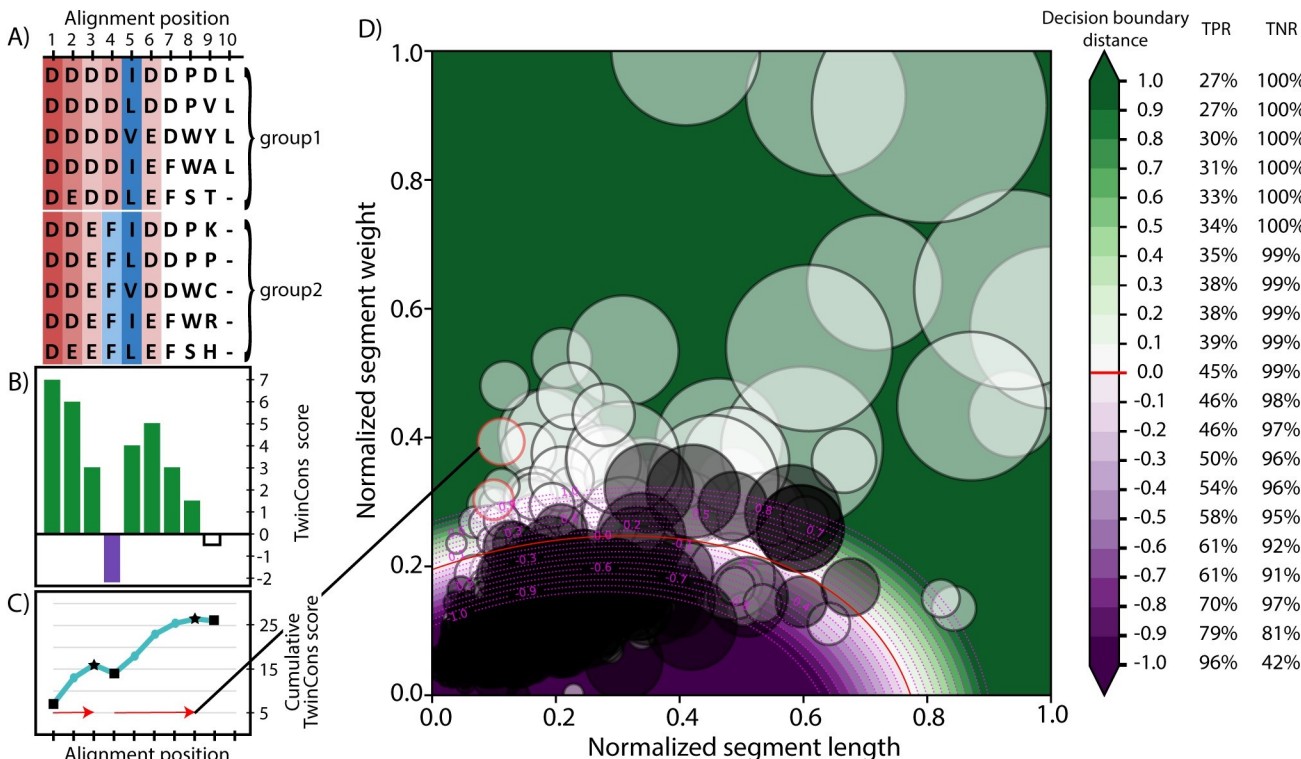

**Fig 1. TwinCons results from composite alignments.** (A) Example composite sequence alignment. Conserved positions within each group are shaded blue for hydrophobic residues and red for charged residues; variable positions are not shaded. Consistent color spanning two groups represents conservation. Change of colors between but not within groups represents signature. (B) TwinCons score for each position in the example composite alignment. Conservation spanning groups produces large positive TwinCons scores (green, positions 1–3, 5). Conservation within but not between groups produces negative scores (purple, position 4). Variability within and between groups produces scores near 0 (white, position 9). Columns with high proportion of gaps produce score equal to 0 (position 10). (C) Cumulative TwinCons score calculated from positional scores. Local minima are indicated with black squares and local maxima are indicated with black stars. Segments are defined as the alignment positions between adjacent local minima (squares) and maxima (stars). Resulting segments are indicated with red arrows. (D) TwinCons results for segments from the BaliBASE multiple alignments. Segments from alignments with related sequence groups are shown with white circles and segments from alignments with unrelated sequence groups are shown with black circles. Each alignment produces multiple segments with normalized length and weight plotted on the 2D graph. One alignment can produce multiple segments (white circles with red boundaries). Segment length and weight is normalized by the maximal length and weight present in the dataset. Circle sizes indicate absolute segment lengths. The decision boundary calculated between segments from related sequence groups (white circles) and segments from unrelated sequence groups (black circles) is shown with red line. Distance away from the decision boundary in the range of -1 to +1 is shown with a diverging gradient and magenta dotted line. Cross validation statistics of true positive rate (TPR) and true negative rate (TNR) for detecting correct assignments of the alignments for each decision boundary distance are on the right.

segments with different lengths. The sum of scores for each position within a segment is considered as the weight of a segment. Thus, there are two variables describing each segment–length and weight; they are normalized against the lengths and weights of the dataset used for evaluation. Alignments that produce positive segments with greater lengths and weights have groups of sequences that are more similar than groups from alignments that generate smaller segments (Fig 1D). An alignment comprised entirely of random sequences will produce Twin-Cons score around zero for all alignment positions when using uniform baseline correction, and the cumulative score will have many troughs. All positive segments will have short lengths. In contrast, an alignment comprised entirely of identical sequences will produce a continually increasing cumulative score, without any troughs, resulting in a single positive uninterrupted segment with length equal to the number of columns within the alignment. Therefore, groups within alignments that produce larger positive segments have sequences that are more similar and are more likely to share ancient ancestry. Detection of segments and calculation of

TwinCons scores can be automated. Below we provide a detailed description of the automation process and the necessary parametrization procedure. Parametrization depends on a large number of composite alignments and makes the inclusion of structure-informed matrices intractable. We describe the rationale for the parameter choices we have made and possible alternatives that have been studied during the parametrization.

**Automated identification of protein segment similarities.**    To determine whether two sequence groups are related we must define a boundary that separates short segments from long segments. The discrimination is performed in Cartesian space of the segment lengths and weights (Fig 1D). To calculate a separation boundary between segments generated from alignments with related and unrelated groups, we generated multiple composite alignment training sets that include alignments from random sequences with INDELible [67] and from biological sequences using rProteins, the BaliBASE database [68], and PROSITE [69,70]. Each set contains true positive composite alignments (TP) with groups known to be ancestrally related and true negative alignments (TN) with groups that do not share ancestry (Table 1). Some TN composite alignments were filtered to ensure that they do not contain groups with shared ancestry. Complete description of the four training datasets is available in the Supplementary text in S1 Appendix.

Using the training datasets, we executed performance optimization and selection of various TwinCons and segment delineation parameters. The parameters were evaluated to discern the combination that best discriminates between TN and TP. Afterwards, we split the training datasets to evaluate the best penalty parameter for training a classifier. The best performing classifier was evaluated against the full datasets to calculate its precision and to define a significance threshold away from the boundary that defines a segment with high similarity between the groups of the composite alignment. The detailed steps of parametrization and training as well as the selected options for each parameter are described below.

**Training a decision boundary.**    TP and TN composite alignments were used as a training set for classifiers based on support vector machines (SVM) [72], or forests of randomized decision trees [73]. An SVM can find an optimal decision boundary that separates the TP and TN parts of a dataset. A random forest of decision trees gives a probabilistic result based on the voting average of each tree. We compared classifiers from the SVM and random forest families generated with the python *sklearn* library [74]. Each of the training datasets was used to generate a separate classifier which were validated against each other. Each segment was represented by two variables–normalized segment length and total segment weight, additionally each segment's importance was weighted by its total length. Every alignment produces segments that are very small and short. The difference between alignments with related (TP) and unrelated

**Table 1. Sources, numbers, and applied filtering of training datasets.** Details for the source of each training dataset are given as reference to relevant publications. Numbers of TP and TN indicate the number of composite alignments used to train classifiers. Details on the composite alignment generation and filtering procedures for each dataset are available in the Supplementary text in S1 Appendix. The INDELible program uses a control file which we provide as S8 Dataset. All composite alignments are available at https://apollo2.chemistry.gatech.edu/TwinConsDatasets/.

| Dataset | Source | TP composite alignments | TN composite alignments | TN Filtering |
|---------|--------|-------------------------|-------------------------|--------------|
| INDELible | Simulator of biological evolution [67] | 22 | 179 | Removed 11 TN for which the random generation performed poorly. |
| Ribosomal proteins | Sparse and Efficient Representation of the Extant Biology (SEREB)[71] | 32 | 178 | Combination of SSU rProteins for TN |
| BaliBASE | BaliBASE multiple alignment suite reference dataset 3 [68] | 38 | 141 | Removed TN, which contain alignment groups with similar structural folds or functions (S1 Dataset). |
| PROSITE | Protein patterns and profiles [69,70] | 120 | 36856 | Combinations based on motifs within the same documentation entry. |

(TN) groups is that alignments with related groups also produce larger and heavier segments. Therefore, when training the decision boundary, we tried using all segments or filtering out the bottom 50% of scoring segments. When filtering each alignment, we used only the largest segments that cover 50% of the total normalized length and total absolute length of all alignment segments.

**Parameterization of TwinCons segment calculation.** Calculations of the TwinCons scores and segments depend on multiple parameters: a) the substitution matrix used, b) the baseline correction (uniform or background frequency), c) the weighing algorithm for sequences, d) threshold for determining segment boundaries (the window size for smoothing the cumulative score or the absolute intensity and length thresholds), e) percentage gaps to be removed from the alignment before a calculation is done, f) filtering out low scoring segments, g) treating signature positions as conserved positions, h) calculating a single average segment position (central mass) from all segments of an alignment, and i) the classifier used.

To identify the combination of parameters that produces most robust differentiation between segments from TP and TN composite alignments, we generated classifiers for segments generated with combinations of TwinCons parameters. Then we calculated a receiver operating characteristic by testing the training sets against the other datasets (Fig A in S1 Appendix and S2 Dataset). Performance can be measured by calculating the percentage area under curve (AUC) statistic.

Our results demonstrate that SVM classifiers consistently outperformed all types of random forest-based classifiers in all combinations of training/testing datasets. SVM classifiers based on the BaliBASE dataset produced best results, often above 95% AUC, against the other datasets. S2 Dataset holds detailed results for all tested parameter permutations.

The parameter combination for BaliBASE classifier that produced most robust differentiation between all datasets was:

a) Substitution matrix: LG

b) Baseline correction with background frequency

c) No weighing of sequences

d) Cumulative segment delineation with smoothing window 9

e) Positions with more than 90% gaps were removed

f) Use the top 50% of segments

g) Do not treat signature positions as conserved

h) Do not calculate central mass of segments

i) Use SVM classifier.

After the selection of parameters of TwinCons and the type of classifier two additional parameters related to the SVM calculation need to be defined. These are the regularization (penalty) parameter and the gamma coefficient of the kernel function.

**Penalty and gamma selection for the SVM classifier.** Determining a decision boundary by using SVM involves the parameters of penalty, kernel, and gamma function for the algorithm. We used the default *sklearn* settings for kernel (rbf). To determine a penalty value, we used a cross-validation methodology where we split a training dataset into 3 folds. This was done by randomly assigning segments from each alignment to one of the three folds, each fold was further split in a training and testing dataset. A classifier was built using the training portion of the fold and tested against the testing portion with different penalty and gamma

parameters. Receiver operating characteristics (ROC) curves were generated for different penalty and gamma values for each fold. Using multiple folds of the training data allowed us to determine standard deviations between runs for the ROC curves. Performance was measured by calculating the AUC statistic. The gamma parameter did not significantly alter the performance of classifiers based on BaliBASE or PROSITE datasets (Fig B in S1 Appendix). Therefore, we use the automatic setting which sets gamma to 1 divided by the number of features. In our case we have two features (segment length and weight), and gamma is 0.5. Different datasets produced different penalties as best performers (Fig B in S1 Appendix). Classifiers built from the BaliBASE, and INDElible datasets showed best results with penalty of 20. Classifiers built from the PROSITE dataset showed improved results with the two lowest tested values for penalty. Classifiers built from the rProtein dataset did not show significant difference across all tested penalty values. We selected to use the penalty of 20, since it gives best results for our best-performing dataset (BaliBASE) (Fig B in S1 Appendix).

**Choice for significance threshold.** Using the calculated decision boundary to directly determine whether a segment has strong similarity between the groups can generate false positives or negatives (Fig 1). We used the AUC performance statistics from the best parameter combination and classifier dataset (BaliBASE) to determine a reasonable threshold away from the decision boundary that lowers false positive results. This threshold is used throughout the manuscript to determine significant segments. Each dataset was tested against the BaliBASE classifier, by calculating TPR and TNR for distances away from the decision boundary from -5 to 5 with a 0.1 step. A distance of 0.7 from the decision boundary produced robust results against the PROSITE (100% TNR, 42% TPR), the INDElible (68% TNR, 100% TPR), and the rProtein (90% TNR, 100% TPR) datasets (S4 Dataset). Setting the significance threshold to a more conservative value of 1.5 would ensure that even the INDElible dataset produces TNR above 90% but would limit the TPR results to 35% in the PROSITE dataset (S4 Dataset). We selected a significance threshold of 0.7 as it gives good results against all datasets from biological sources.

**Probability estimation.** An SVM classifier is not intrinsically a probabilistic classifier, however it can output a probability for the prediction it makes. We include this probability in our outputs for user convenience. In our tests all segments with distance greater than 0.7 from the decision boundary have a probability greater than 90%. Furthermore, we provide scripts to generate calibrated SVM classifiers from our training data, which should behave analogous to probabilistic classifiers.

**Query alignments.** Here we probe the ability of TwinCons to detect highly conserved sequence motifs for 27 proteins or protein fragments from the translation and transcription systems with known or suspected ancestry (Table D in S1 Appendix) based on structural similarity or previous literature [75,76] (S5 Dataset). All translation proteins were retrieved from the advanced visualization website ProteoVision [77]. Sequences for transcription systems were retrieved from NCBI and are available at https://apollo2.chemistry.gatech.edu/TwinConsDatasets/. Alignment of Tyrosyl and Tryptophanyl aminoacyl tRNA-synthetases were retrieved from Fournier and Alm [78]. To compute TwinCons, we generated composite alignments for each pair of candidates.

**Querying an alignment against trained classifier.** To detect significant segments within a single alignment we calculate TwinCons and segments with the same parameters used for the classifier. We calculate the distance from the decision boundary for each calculated alignment segment. If the distance is greater than 0.7 the segment is considered as a significant hit. TwinCons reports the significant segments, their probability, their distance from the decision boundary, and the alignment ranges they span in a csv format.

**Evaluating TwinCons performance against HHalign on training datasets.** To evaluate the performance of the TwinCons segment calculation we compared it to HHalign 3.0.3 [31]. HHalign can take as input a template alignment and a query alignment to produce a combined alignment. All pairs of alignments in the rProteins, INDELible, BaliBASE, and PROSITE datasets were aligned with HHalign with default parameters, varying only the parameter that filters input alignment columns by percentage of gaps. By default, HHalign generates an output alignment that includes segments of local similarity between two input alignments. HHalign also calculates scores for the output alignment including P-value, E-value, and probability of homology of the locally aligned segment. We used the HHalign E-value as a threshold to generate ROC curves for a range of E-values starting from 0 and ending at 1 with a step of 0.001. HHalign performed perfectly (100% AUC) in detecting homologous and non-homologous groups in the rProtein and INDELible datasets (Fig C in S1 Appendix). HHalign performed almost perfectly on average in detecting homology in the BaliBASE and PROSITE datasets (97% and 92% AUC). The exclusion of columns by different percentage of gaps showed weak influence on the results with removal of 10% showing the lowest AUC (Fig C in S1 Appendix). While TwinCons was not designed to detect homologies, it is able to reach 90% AUC for the BaliBASE (Fig B in S1 Appendix) and PROSITE (S2 and S4 Datasets) datasets and 100% for the rProtein and INDELible datasets, performing nearly as well as HHalign (Fig C in S1 Appendix).

## Results

### TwinCons: Score that highlights conserved and signature regions

TwinCons detects highly conserved and signature sequences between two groups within a composite alignment. Here, we demonstrate the utility of TwinCons using ribosomal protein uL2. Archaeal uL2 sequences form one group in the alignment and bacterial uL2 sequences form the other group. In addition, we analyze caspases and metacaspases, which are cysteine proteases thought to share ancestry. For the caspase and metacaspase composite alignment, caspases form the first group and metacaspases form the second group. We provide a detailed comparison of TwinCons results with two previously described conservation metrics ConSurf [33–35] and Zebra2 [17,18]. ConSurf can detect conservation within proteins and RNA across an entire alignment column, while Zebra2 detects signature residues in a composite alignment. We calculate protein conservation of the common composite alignments with TwinCons, ConSurf and Zebra2 and compare their scores directly (S10 Dataset), we further map them on 3D structures (Figs 2 and 3, and D in S1 Appendix).

**TwinCons detects highly conserved and signature positions in composite alignments.** TwinCons readily detects sequence conservation in ancient proteins. uL2 is thought to be one of the oldest ribosomal proteins [81–83], and its high sequence conservation across phylogeny has been documented [81]. High conservation is revealed within the globular region as well as the extended loop of uL2 by both TwinCons and ConSurf (dark green, Fig 2). The results from TwinCons are consistent with those from ConSurf for the sites that are highly conserved across the entire alignment (Fig 2 and S10 Dataset).

In addition to detecting the conserved positions within a composite alignment, TwinCons identifies signature sequences that are conserved within each group and are different between the groups. In that way TwinCons highlights important evolutionary differences between archaeal and bacterial sequences (Fig 2B). Such sites are indicated by negative TwinCons scores. Thus, sites 41, 49, 150, 171, 199, 202, 206, 212, 248 in uL2 (*E. coli* numbering scheme) have large negative TwinCons scores. These signature sites cover bacterial and archaeal structural regions that superimpose well (Fig 2). Since these residues are conserved within each

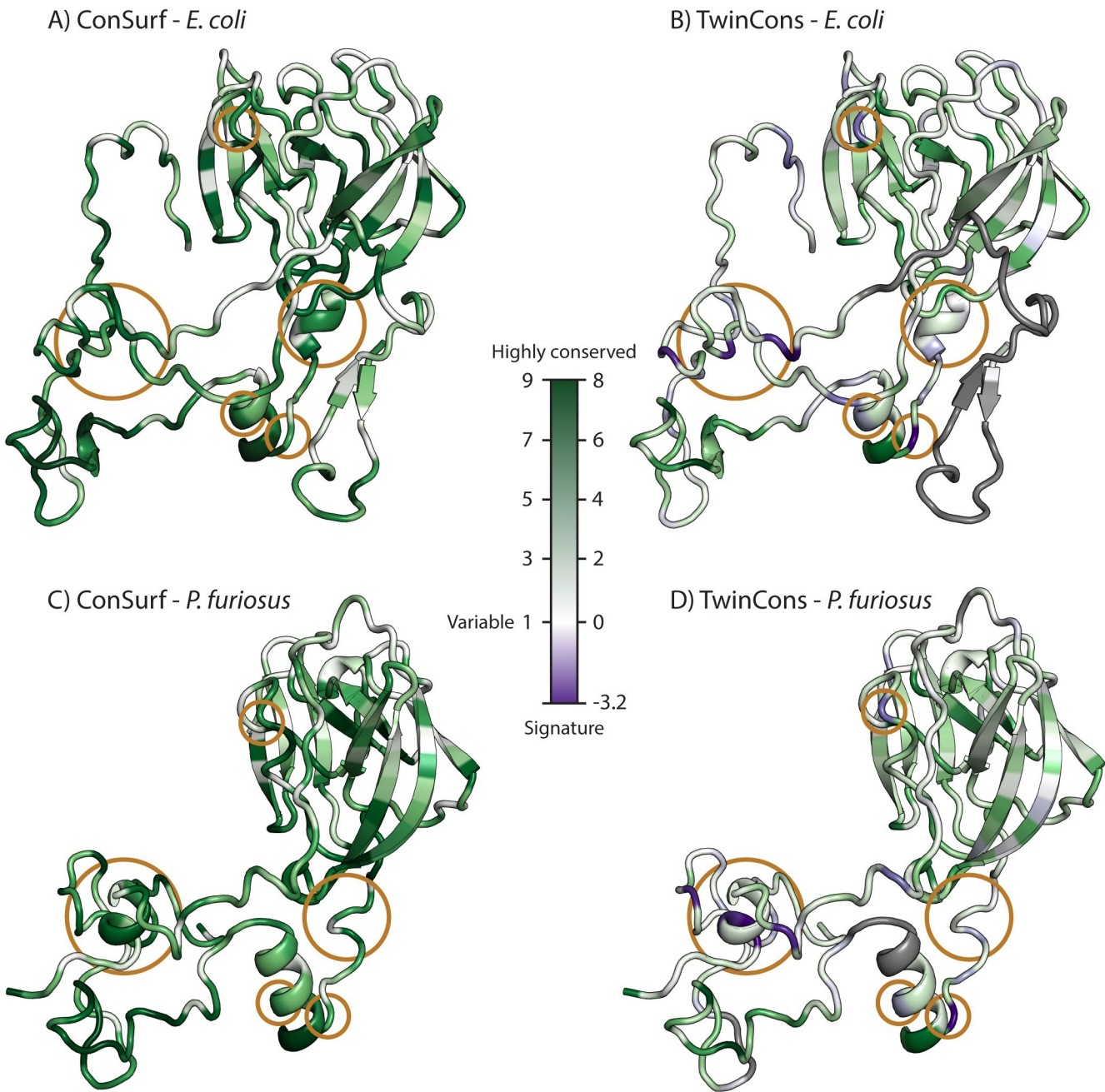

**Fig 2. Comparison of ConSurf and TwinCons results by structural mapping.** (A) ConSurf results from a composite sequence alignment for uL2 of archaeal and bacterial sequences mapped on the *E. coli* uL2 structure from PDB ID 4V9D [79]. (B) TwinCons results for the same composite alignment. (C) ConSurf results mapped on the *P. furiosus* uL2 structure from PDB ID 4V6U [80]. (D) TwinCons results mapped in the same way. White indicates positions with highly variable sequences across the alignment. Purple indicates signature positions. Gray indicates heavily gapped regions. Orange circles highlight regions of difference between TwinCons and ConSurf. The default coloring scheme of ConSurf was adjusted to a gradient from white to dark green to allow correct comparison between ConSurf and TwinCons.

branch of the TOL but have different identities between the branches, TwinCons can point to sequence changes with evolutionary significance. By contrast, ConSurf indicates these sites are moderately conserved (Fig 2, orange circles).

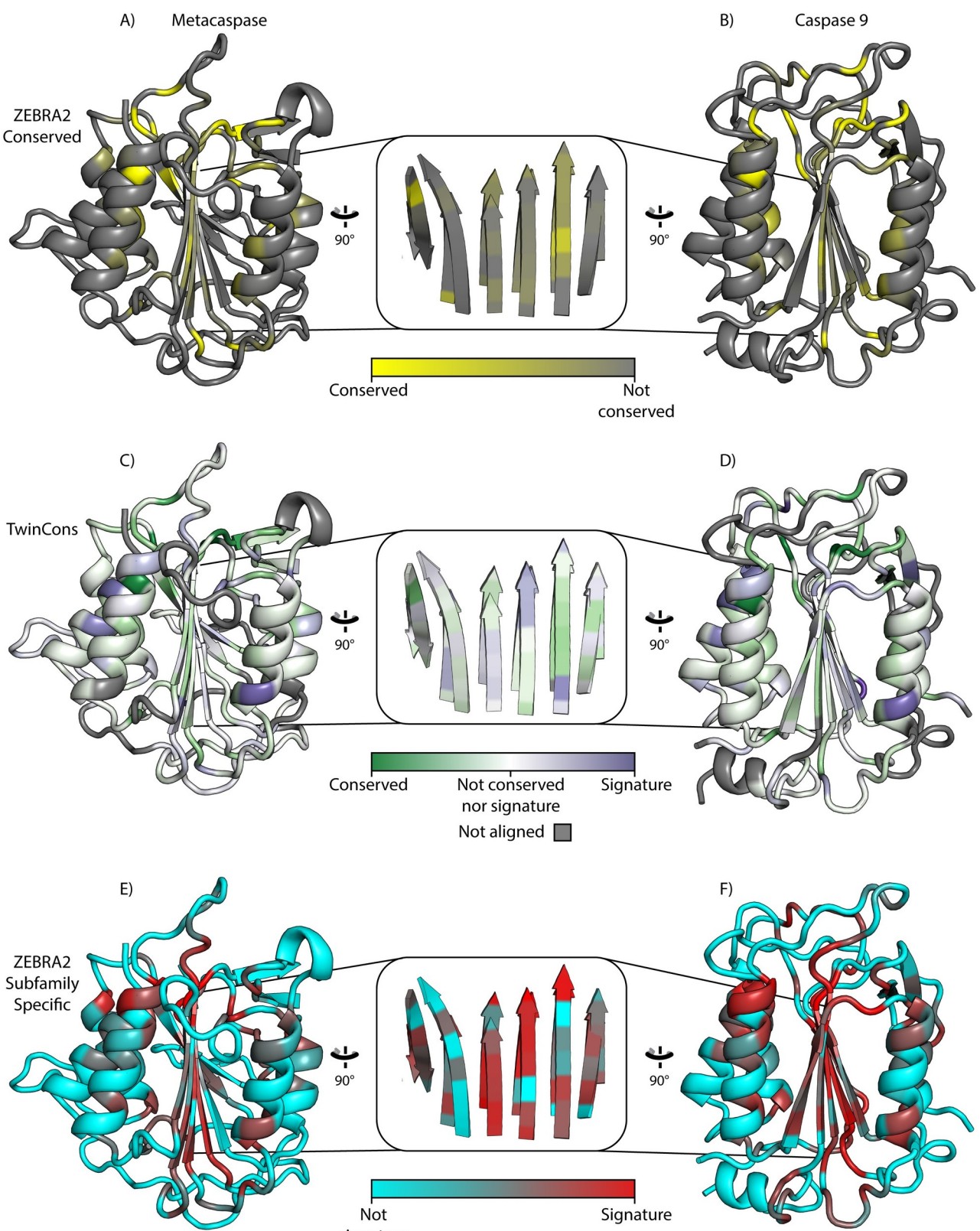

**Fig 3. TwinCons is a convenient tool to study conservation and signatures.** (A) Zebra2 conservation mapped on yeast metacaspase (PDB ID 4F6O [86]). (B) Zebra2 conservation mapped onto human caspase-9 (PDB ID 1JXQ [85]). (C) TwinCons conservation mapped on yeast metacaspase. (D)

TwinCons conservation mapped onto human caspase 9. (E) Zebra2 signatures (subfamily specific) mapped on the metacaspase (F) Zebra2 signatures (subfamily specific) mapped on caspase 9. Zebra2 conservations are shown with a gradient from yellow to gray, with yellow indicating highly conserved and gray indicating non-conserved residues. Zebra2 signatures are shown with a gradient from cyan to red, with red indicating signatures. TwinCons conservation is shown in white to green gradient (green is more conserved). TwinCons signatures are shown with a white to purple gradient (purple indicates signatures). For TwinCons, white indicates non-conserved and gray indicates alignment positions that have sequences in only one group. The inset shows the superimposition of the central β-sheets of each pair of proteins.

TwinCons, like ConSurf, detects heavily gapped alignment positions. TwinCons can discriminate between alignment positions that contain gaps only within one group and alignment positions with gaps spanning the entire column. For example, the N-terminal β-hairpin of uL2 (1–37 *E. coli* numbering) is a feature present only in bacterial ribosomes and it should not exhibit any TwinCons conservation, for that reason it lacks a score and is colored gray (Fig 2B).

**TwinCons detects signature-rich regions.   TwinCons can highlight alternative sequences of conserved structures.** TwinCons can identify structurally similar regions with high frequency of signature columns within a composite alignment. For example, a short α-helix, located between positions 196 and 206 of uL2 in *E. coli*, displays seven highly conserved sites when calculating its evolutionary rate with ConSurf (Fig 2A and Table C in S1 Appendix). In contrast, TwinCons reveals four signature sites for Bacteria and Archaea in the same region. Three of these positions coincide with the conserved residues detected by ConSurf (Fig 2B and Table C in S1 Appendix). Analysis of the sequence similarity within each (archaeal and bacterial) group reveals that consensus sequences calculated for each group are distinct (Table C in S1 Appendix). Yet, the 3D structures of archaeal and bacterial uL2 exhibit a common α-helical element in this region (Fig E in S1 Appendix), therefore TwinCons can be used to detect alternative sequences for common structural elements.

**TwinCons provides convenient representation of signatures and conservation.** TwinCons is robust and can probe evolutionary relationships between distantly related paralogs. Cysteine-dependent aspartate-directed proteases called Caspases regulate apoptosis in metazoans. Similar apoptotic proteases, called metacaspases, are found in plants, fungi, and unicellular organisms. Caspase and metacaspase structures share α/β three-layered sandwich architecture characterized by a single centrally positioned parallel β-sheet and α-helices on both of its sides; their evolutionary relationship predates the last eukaryotic common ancestor [84]. Here we demonstrate the use of structurally informed substitution matrices in TwinCons to compare sequence similarity of caspases and metacaspases. We cross check the TwinCons analysis with Zebra2 results computed from the same alignment. We map these results on structures from a caspase (PDB ID: 1JXQ [85]) and a metacaspase (PDB ID: 4F6O [86]) (Fig 3).

The two groups of proteases are evolutionary related [84] and structures representing each group superimpose with an RMSD of 2.9 Å. TwinCons highlights four distinct types of sites; i) universally conserved sites associated with the catalytic region; ii) moderately conserved buried residues; iii) highly variable solvent exposed residues; iv) signature residues at the periphery of secondary structural elements (Fig 3D–3F). Zebra2 detects all 4 types of regions by using separate scores for conserved (Fig 3A–3C) and signature residues (Fig 3G–3I). The two Zebra2 scores are not mutually exclusive. Zebra2 detects more signatures within the alignment than TwinCons does, and just within the core β-sheet there are 10 Zebra2 signature sites (Fig 3H). TwinCons detects fewer signature residues in the entire alignment and 8 within the core β-sheet (Fig 3E). The difference between the two scores stems from low penalties for signatures in the structural matrices used by TwinCons. While Zebra2 requires two separate mappings to visualize its scores (Fig 3A–3C and 3G–3I), TwinCons produces a single score that detects highly conserved, variable and signature residues, which are mapped on the structure (Fig 3D–3F).

### TwinCons: Score that detects sequence similarity between pairs of proteins

The simplicity and universality of TwinCons in analyzing local differences within protein composite alignments were leveraged to detect highly similar regions for any two proteins. We developed an automated parsing protocol aimed to detect continuous stretches of columns with high TwinCons across the entire length of a composite alignment. We refer to such regions as segments. A segment spans a range of alignment positions with a continuously increasing cumulative TwinCons score after a smoothing with window size 9 (suggested by TwinCons parameterization protocol) has been performed. We used a supervised learning model to predict whether a composite alignment contains regions with greater TwinCons scores compared to those from composite alignments that include ancestrally unrelated groups. To explore the ability of TwinCons to detect highly conserved sequence motifs, we applied this method to 27 proteins or protein fragments from the translation and transcription systems with known or suspected ancestry (Table D in S1 Appendix and S5 Dataset) [75,76]. The results from 36 composite alignments (some protein pairs have more than one alignment) are summarized in Fig 4.

Automated TwinCons analysis reveals similar segments in 10 pairs of proteins from 13 composite alignments. TwinCons detects highly similar segments between elongation factor Tu (EF-Tu) and Initiation factors 2 (IF2) and 5 (aIF5 –archaeal variant of IF2), between rProteins aL8, aL30, and eS12, between ribosomal protein bS1 and RNAP subunit 7, between rProteins aL37 and bL34, between rProteins aL14 and eL27. TwinCons further detects more marginal similarities between rProteins aL42 and bL33, between Tyrosyl and Tryptophanyl aminoacyl tRNA-synthetases, and between rProteins uL2 and bL34 (Fig 4 and Table 2). We describe TwinCons results in terms of number of segments, significance, and the total length of the detected segments. Complete descriptions and 3D maps of the highly similar segments are available in the Supplementary text and Table E in S1 Appendix. We further provide a comparison of our results with HHalign [31,87], an established method for detecting homology in protein sequences using hidden Markov's algorithms.

Results between HHalign and TwinCons from the query set are largely similar (Table 2), except for alignments that include multi-domain proteins (EFTu-aIF5 and bS1-aRNAP7), for which TwinCons yields shorter ranges. TwinCons segments are generally more conservative and fragmented than HHalign segments (Table 2; aL8-aL30, aL8-eS12, aL30-eS12). TwinCons detects similarity between shorter segments more readily than HHalign (Table 2; bL34-aL37). TwinCons detects at least a single segment for alignment pairs where HHalign has high confidence results (S9 Dataset). To illustrate TwinCons detection of a possible similarity for a marginal case, we provide detailed description of the automated TwinCons results for a pair of proteins related by a circular permutation.

**Sequence similarity between the circularly permuted bL33 and aL42.** TwinCons detects a conservation signal within a β-hairpin when comparing rProteins bL33 and aL42. Previous work demonstrated the structural relationship between rProteins bL33 and aL42 [88,89]. These proteins are colocalized in bacterial and archaeal ribosomes and have very similar structures, related by a circular permutation. Here we refer to them with their classical domain names bL33 and aL42 [90] for clarity, however we and others have proposed that they should share the name uL33 [88,91]. We probed whether TwinCons can detect the ancestral relationship within the composite alignment of bL33 and aL42. To do that we aligned one composite alignment with unpermuted sequences (bL33-aL42) and two composite alignments containing either a permutation of bL33 (bL33$_{CP}$-aL42) or of aL42 (bL33-aL42$_{CP}$).

In both permuted composite alignments TwinCons detects a pair of segments with sequence similarity between bL33 and aL42 (Fig 4 pink and Table 2 and S9 Dataset). One of these segments is significant and it corresponds to the N-terminal β-hairpin of the aL42 structure (PDB ID: 4V6U

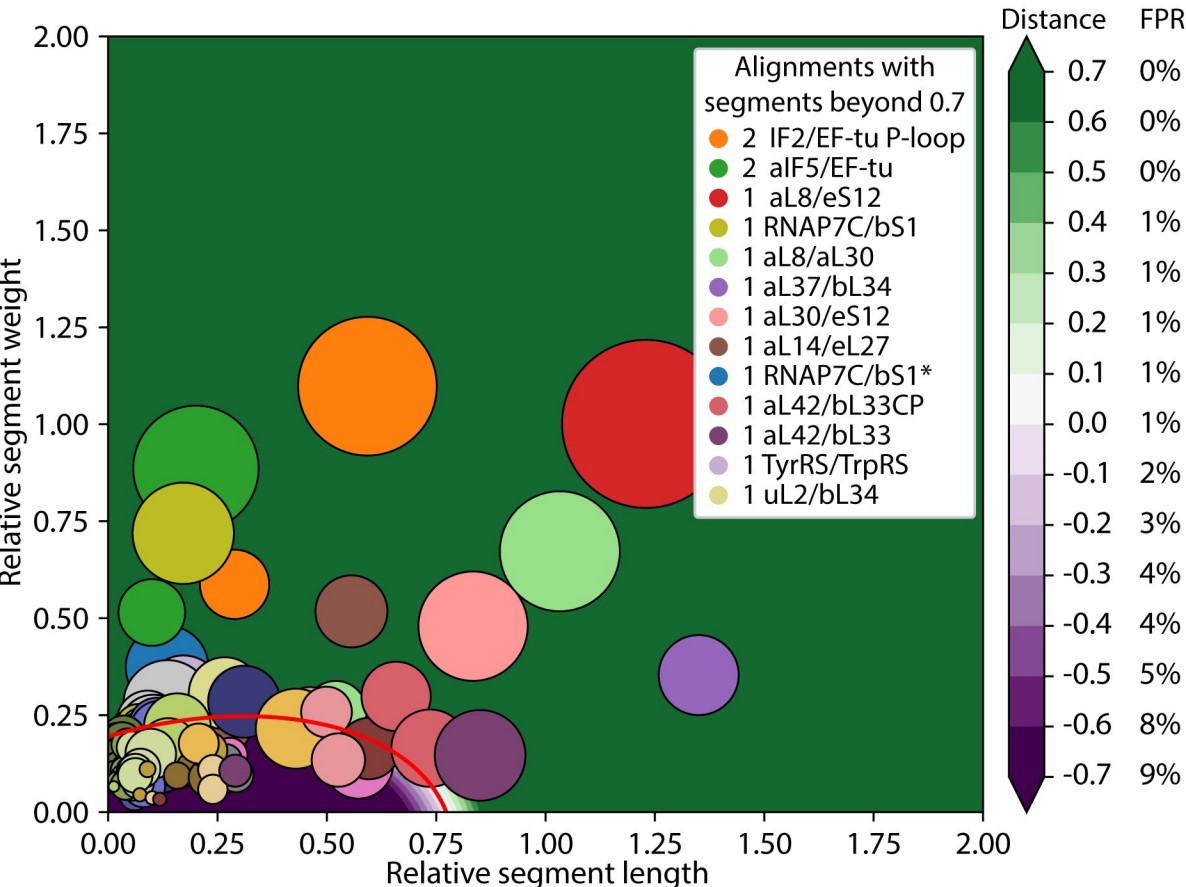

**Fig 4. TwinCons can detect noisy sequence similarity.** Results from composite alignments of genes related by putative ancient duplication events. Segments are represented with uniquely colored circles for each alignment. Circle sizes indicate segment length. The segment weight and length are normalized relative to the training BaliBASE dataset. The decision boundary calculated from the BaliBASE dataset is shown with a red line, distances from the boundary are shown with a diverging gradient (purple to green). The inset shows the number of segments for each alignment at a significant distance from the decision boundary. Negative distance values indicate positions below the decision boundary and positive values indicate positions above the boundary. Alignments with segments close to the decision boundary are at the bottom of the legend and those further away are at the top. On the right, each distance from the decision boundary is associated with the calculated false positive rate for the BaliBASE dataset. *Structure guided alignment.

[80]) and the C-terminal β-hairpin of the bL33 structure (PDB ID: 4V9D [79]) (Fig F in S1 Appendix). These two β-hairpins superimpose well and are structurally similar (Fig F in S1 Appendix). The second segment has a weaker TwinCons signal and corresponds to the C-terminal β-hairpin of the aL42 structure and the N-terminal β-hairpin of the bL33 structure (S9 Dataset). HHalign confirms the circular permutation over the entire range in both composite permuted alignments (Table 2 and S9 Dataset). Thus, our results confirm a previous proposal on common ancestry of these two proteins and suggest that they should be treated as a universal protein uL33 [88].

We also note that both TwinCons and HHalign detect a short region of marginal similarity (Table 2) in a non-permuted alignment of bL33 and aL42 (Fig 4 violet and S9 Dataset). These results appear to be low-scoring artifacts of a structurally inconsistent sequence alignment.

## TwinCons results between archaeal and bacterial rRNA

TwinCons is a useful metric to study conservation and ancestry of nucleic acids. For example, TwinCons can be applied to identify signature positions in rRNAs grouped into the three

**Table 2. Significant segments detected by TwinCons for protein composite alignments.** Sequence ranges for segments detected as significant with TwinCons and in parenthesis their respective distance from the decision boundary. Sum of all significant segment lengths for each alignment. HHalign E-value confidence results of aligning the group sequences. Number of aligned columns with HHalign; for each pair, each protein was used as template and the higher result is reported here. Figures that show detected segments in 3D are indicated under the alignment names. Indexing species for the indicated ranges and complete data for all query alignments are available in S9 Dataset.

| Composite alignment | TWC Segment residue ranges (Boundary dist.; prob.) | Total length | HHalign E-value | HH Range | HHalign aligned columns |
|---|---|---|---|---|---|
| IF2/EF-Tu P-loop | 10–35 (5.4; 100%)<br>95–145 (12.8; 100%) | 75 | $2.8e^{-29}$ | 16–181 | 140 |
| aIF5/EF-Tu (Fig J in S1 Appendix) | 8–32 (4.7; 100%)<br>95–141 (11.1; 100%) | 70 | $3.6e^{-22}$ | 15–274 | 237 |
| aL8/eS12 | 30–93 (10; 100%) | 63 | $5.1e^{-18}$ | 22–137 | 115 |
| RNAP7C/bS1 (Fig K in S1 Appendix) | 83–120 (8.2; 100%) | 37 | $3.6e^{-15}$ | 114–203 | 72 |
| aL8-aL30 | 22–66 (6.8; 100%) | 44 | $4.2e^{-15}$ | 32–124 | 91 |
| aL37-bL34 (Fig L in S1 Appendix) | 1–26 (5.7; 100%) | 25 | 0.39 | 51–51 | 1 |
| aL30-eS12 | 3–43 (4.1; 100%) | 40 | $7e^{-8}$ | 20–114 | 92 |
| aL14-eL27 | 4–30 (4.7; 100%) | 26 | $6.5e^{-9}$ | 3–66 | 64 |
| RNAP7C/bS1 Struct. based | 83–114 (2.1; 99%) | 31 | $1.2e^{-18}$ | 120–286 | 136 |
| aL42/bL33CP | 4–26 (1.3; 95%) | 22 | $3.1e^{-8}$ | 76–154 | 68 |
| aL42/bL33 | 3–31 (1.2; 93%) | 28 | $5.8e^{-5}$ | 101–122 | 20 |
| TyrRS/TrpRS | 38–63 (0.9; 89%) | 25 | $3.5e^{-28}$ | 118–287 | 151 |
| uL2/bL34 | 13–39 (0.81; 87%) | 26 | 0.064 | 77–107 | 30 |

major domains of life [11], provided that the transformation matrix in Eq 2 is adjusted for nucleic acids. Previously, Woese et al. have detected signature positions within rRNA sequence and structure [12,92]. Here, we used TwinCons computed for sequence alignments of bacterial and archaeal rRNAs to identify ancient signature nucleotides.

We mapped the values of TwinCons onto a representative LSU 2D and 3D structures for each group–*E. coli* for Bacteria (Fig 5) and *P. furiosus* for Archaea (Fig G in S1 Appendix) [12,92]. We used TwinCons to identify conserved, variable and signature positions between bacterial and archaeal rRNA sequences (Supplementary text in S1 Appendix). Regions of high functional importance for the ribosome, such as the exit tunnel, the central protuberance, the central pseudoknot (CPK), the peptidyl transferase center (PTC) and the sarcin-ricin loop, are highly conserved and have high TwinCons scores (dark green regions in Figs 5 and G in S1 Appendix). Regions on the ribosomal surface, known to contain multiple long insertions with highly variable sequences, are correctly detected as either highly variable (white regions in Figs 5 and G in S1 Appendix) or heavily gapped (gray regions in Figs 5 and G in S1 Appendix).

Signature sites detected by TwinCons are sparse and are distributed throughout the rRNA structure. Surprisingly, signature sites are not confined to the variable ribosomal surface regions (Fig 5C and 5D and G in S1 Appendix). Signatures are found both in conserved regions of functional importance, like the central pseudoknot in the SSU (helices 1–3 and 28 in Fig 5A) and the PTC in the LSU (Helices 73 and 89–93 in Fig 5B), signature sites are also found in variable regions like helix 33 (Fig 5A) and Helix 55 (Fig 5B). In total we detect 208 signature nucleotides in the SSU and 294 signatures in the LSU (S7 Dataset). Some signatures occur at base-pairing nucleotides, revealing highly conserved base pairs within each group which co-vary between the two groups (helix 30 in Fig 5A and Helix 69 in Fig 5B). By identifying the signature positions in rRNAs, we uncover the deepest evolutionary changes in the translational machinery that define the phylogenetic split at the last universal common ancestor.

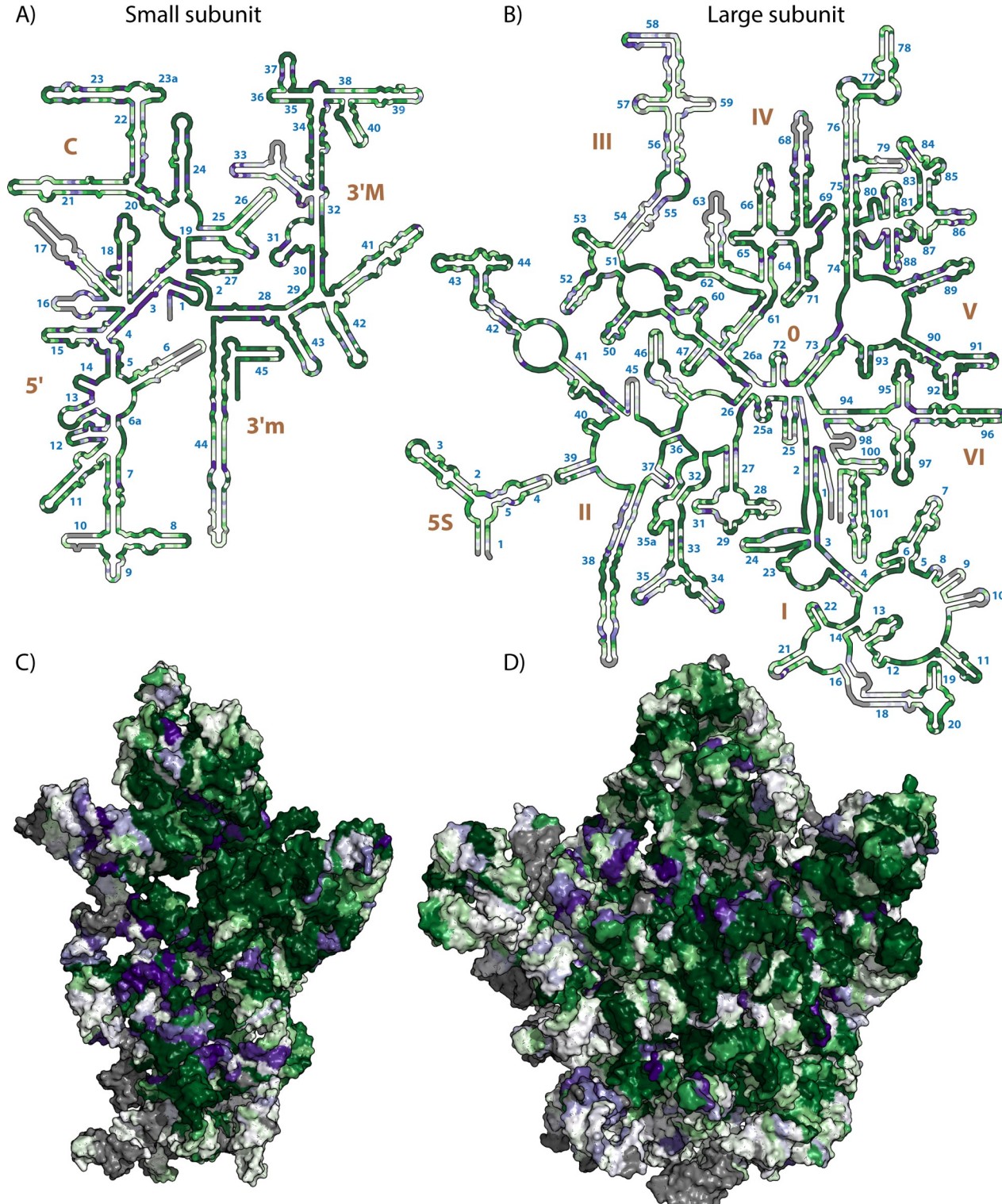

**Fig 5. TwinCons between archaeal and bacterial rRNA sequences, mapped on 2D and 3D representation of the E. coli ribosomal structure.** (A) Secondary structure of the *E. coli* 16S rRNA. (B) secondary structure of the *E. coli* 5S and 23S rRNAs. (C) Surface representation of the three-dimensional structure of the 16S rRNA of the *E. coli* ribosome. D) Surface representation of the three-dimensional structure of the 5S and 23S rRNAs of the *E. coli* ribosome. The 3D structures are viewed into the subunit interface. Gray indicates heavily gapped regions, indicating elements that are present in either bacterial or archaeal rRNAs; dark green indicates highly conserved regions within both bacterial and archaeal sequences; dark

purple indicates signature regions between bacterial and archaeal sequences; white indicates sequence variable regions. Helical numbers are blue and domain numbers are brown. 2D representations were generated with RiboVision [93], and 3D representations were generated with PyMOL [94]. PDB ID used for 3D representations is 4V9D [79].

**TwinCons can identify colocalized patterns in rRNA and rProteins.** To extend our analysis beyond signals obtained from individual sequences we performed a combined sequence and structural analysis of rRNA and rProteins. This analysis reveals colocalized changes suggesting coevolution of the two types of ribosomal biopolymers. Previously Roberts and co-authors correlated the structural signatures between rProteins and rRNA, however they did not identify rProtein sequence signatures or correlate them to rRNA sequence signatures [92]. rProtein uL33 (bL33 or aL42) is located near the E-site of the LSU. Using TwinCons we detected a sequence segment that is highly conserved between Bacteria and Archaea (cyan cartoon Figs 6 and F in S1 Appendix and Table 2). This segment interacts with highly conserved rRNA Helices 82 and 86 (*E. coli* numbering), part of the central protuberance (Fig 6). The other β-hairpin of uL33, which has low sequence similarity between bacteria and archaea, interacts with multiple rRNA signature sites in Helices 81 and 88 (gray cartoon Figs 6 and 5). The α-helical insertion present only in archaeal uL33, also interacts with rRNA Helices 81 and 88 (*E. coli* numbering) (gray cartoon Fig 6). These results indicate correlation between changes in rProtein and rRNA in the most ancient ribosomal regions.

## Discussion

Multiple sequence alignments can provide us with archives of molecular history [95]. These archives are imperfect records of past events; with lost sentences, paragraphs, pages and chapters. These imperfections reflect a complex history of evolution, during which simple point mutations accumulate together with more complex processes of large insertions, deletions, duplications, and permutations. Molecular history has been tracked successfully using protein domains [75,96,97], motifs [57,98,99], and short segments of proteins [58,100]. Comparing two alignment groups within a composite alignment of two genes that are possibly related, and combining sequence and structural analysis, can help unveil the origins and deep history of ancestral genes, pushing the boundaries of what we can learn about the distant molecular past. In general, from a composite alignment, multiple types of patterns can be recovered, revealing distinctive aspects of molecular history. Important patterns include:

- Highly conserved sequence regions across groups highlight critical elements responsible for structural and functional identity of a particular gene.

- Highly variable regions across groups, represent an archive of recent evolutionary history and are essential for fine-grained phylogenetic reconstructions.

- Regions that are conserved within alignment groups but differ between groups (signatures), highlight functional diversification between groups.

Using pre-generated composite alignments, TwinCons detects conserved, variable and signature positions between alignment groups in a single step. TwinCons can be mapped onto three dimensional structures and employs a wide variety of sequence- and structure-informed substitution matrices, as well as sequence weighing based on global similarity measures. Automated visualization of TwinCons has been implemented in ProteoVision [77], an online web-server that enables simultaneous structural visualization and custom data mapping at levels of primary, secondary, and three-dimensional structure of proteins. TwinCons displays signatures and conservation in a single convenient layout and highlights alternative sequences of

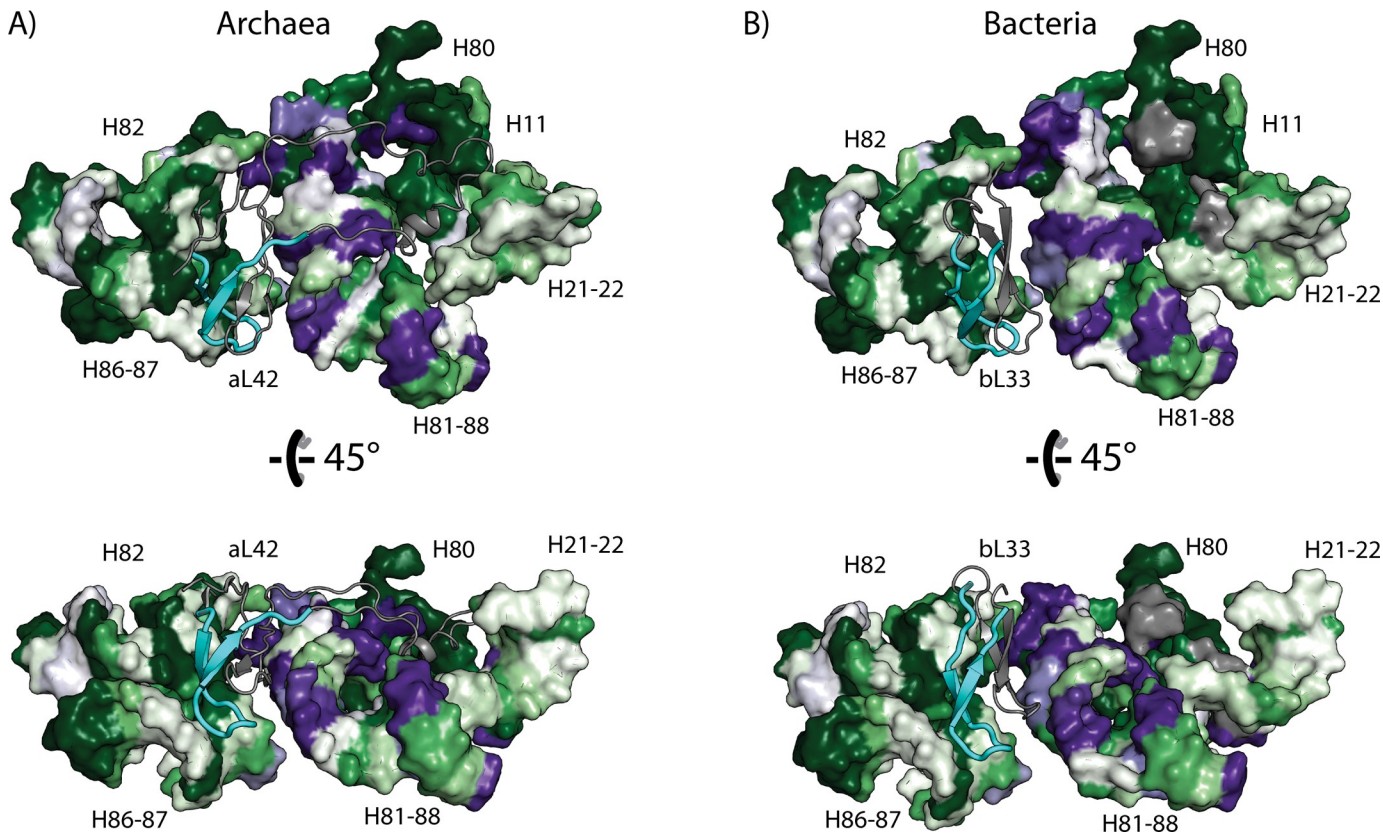

**Fig 6. Correlation between signatures and conservation in archaeal and bacterial rRNA and rProteins detected with TwinCons.** (A) Archaea (rProtein aL42 in *P. furiosus* ribosome). (B) Bacteria (rProtein bL33 in *E. coli* ribosome). rRNA helices are labeled with capital H and canonical numbering. rProteins bL33 and aL42 are shown with cartoon with the highly conserved segment colored in cyan and the rest of the protein colored in gray. This figure was generated with PyMOL [94]. The PDB ID used for panel (A) is 4V6U [80] and for panel (B) is 4V9D [79].

conserved structures. TwinCons identifies structural locations of signatures and conserved residues associated with catalytic regions of distantly related enzymes. Furthermore, TwinCons finds similar segments within composite alignments. TwinCons is generalizable for both protein and nucleotide alignments, facilitating joint analysis between RNA and proteins in complex RNA-protein assemblies such as the ribosome, unveiling ancient molecular history.

## TwinCons highlights structurally important sites in enzymes

Evolutionary related caspases and metacaspases [84] were analyzed here by TwinCons, highlighting four categories of information: i) universal conservation, ii) moderate conservation, iii) high variability, and iv) signatures (Fig 3D–3F). These TwinCons results are consistent with caspase and metacaspase structures.

i. Universally conserved sites of caspases and metacaspases are associated with catalytic regions, signifying conserved catalytic mechanisms.

ii. Moderately conserved sites are found at buried residues, implicating residues that preserve the structural core.

iii. Highly variable sites are found at solvent exposed residues, which do not influence the structure or function of the enzyme.

iv. Signature sites are often observed at the periphery of secondary structural elements, signifying differences in caspase and metacaspase mechanisms of association between secondary structural elements and the structural core.

The comparison of caspases and metacaspases illustrates how TwinCons can quickly highlight conserved, variable, and signature sites with a single score. In that way TwinCons is a useful tool for comparative evolutionary studies between related groups of proteins.

## TwinCons compared to alternative scores

Conservation within sequence alignments is a cornerstone of evolutionary analysis. A wide variety of full column conservation scores have been developed [34,36,52,101]. These conservation scores treat an alignment as a single object (without partitioning into groups) and thus, they cannot discriminate between signature and conserved regions, as exemplified by comparison with ConSurf (Fig 2). One of the first forays in identifying functional protein residues likens the entire protein sequence to a vector [102]. TwinCons also utilizes such notation, when frequencies of residues or nucleotides in each group are represented as vectors and transformed by a substitution matrix. Thus, the calculation of our score is analogous to computation of the energy of a dipole vector in an external anisotropic electric field, defined by an anisotropic tensor. Several statistical methods focus on identifying functionally important regions within a composite alignment [103,104]; these methods lack the ability to incorporate structural data and are protein specific. Toolkits like Diverge3 [54] and Zebra2 [18] that incorporate structural data and map scores on structures have also been developed [105,106]. By contrast, TwinCons provides a single metric that specifies between-group conservation. TwinCons improves on existing signature-identifying methods by providing greater accessibility and flexibility in usage, in calculation methods, in visualization capabilities, and in simultaneous application for RNA and protein. Below, we briefly compare TwinCons with the scores mentioned above and highlight the strengths of each method.

**Comparison of TwinCons with DIVERGE3.**  Diverge3 reveals site-specific divergences (signatures) within a protein family for a supplied alignment and sequence groups. Diverge3 evaluates a Bayesian profile for each group and measures how amino acid residues contribute to signatures. Diverge3 can map its divergence score onto a 3D structure. The divergence statistics employed by Diverge3 are based on analysis of evolutionary rates inferred from a given alignment. Diverge3 is not intended to detect conserved and highly variable regions. Unlike Diverge3, TwinCons uses statistics built into substitution matrices (see below) and provides instantaneous computations of the scores for every position within a composite alignment.

**Comparison of TwinCons signature positions with Zebra2.**  Zebra2 automatically partitions an alignment into multiple pairs of groups and assigns significance values for every grouping. For every detected group, it calculates two separate metrics for conservation and signatures. Zebra2 provides 3D structural mappings, and a sequence similarity network representation of the results [17,18]. TwinCons uses either pre-defined or automated phylogeny-based partitioning of an alignment to define its groups; it computes a single score that distinguishes between conserved, variable and signature regions. The single TwinCons score facilitates visualization and identification of each of these regions and does not require mapping data multiple times, which would be necessary for Zebra2 (Fig 3). The results for signature regions obtained by Zebra2 and TwinCons are consistent, if TwinCons is computed using structure-informed substitution matrices (Figs 3 and H in S1 Appendix) [43].

## TwinCons requires a pre-computed composite alignment

TwinCons is an evolutionary metric computed for each position within the composite alignment. It does not optimize or otherwise alter the input alignment. Thus, TwinCons results are highly dependent on the alignment and the initial determination of sequence groups within that alignment. TwinCons will produce a result for any supplied composite alignment, manually curated or automatic. A given set of sequences, organized in different groups or in different alignments, can produce differing TwinCons scores. Thus, supplied composite alignments and the groupings within them should be carefully checked in advance for sufficient quality.

**TwinCons detects short highly similar segments.** HHalign from the HH-suite [31] can compute an optimal pairwise alignment between HMM profiles generated from two alignments [30,87]. In that way HHalign can produce predictions about protein homology and can identify highly similar regions between two alignments or profiles. TwinCons measures the conservation between two pre-defined groups within an already constructed composite alignment and can identify segments that have higher levels of similarity than observed in datasets of unrelated alignments. The two methods are different, yet their results, from our query set, are largely similar (Table 2). While the overall ranges of TwinCons segments are more conservative and more fragmented than HHalign ranges, TwinCons detects similarity between short segments, which HHalign does not. Thus, TwinCons is more useful for identification of short motifs. This difference is likely the result of the segment delineation rule, used by TwinCons. In future releases of TwinCons we plan to improve on this rule using a probabilistic method that accounts for global similarities.

## TwinCons detects shared segments within the universal core of life

Reuse of small protein segments is ubiquitous in protein evolution. Opportunistic reuse has been observed so often as to raise questions about the primacy of the protein domain as the "atomic unit" of evolution [57,58,98]. The translation and transcription systems contribute to most of the universal gene set of life [71,107,108]. TwinCons identified segments of peptides with high similarity between 9 pairs of proteins from 12 composite alignments, spanning the translational and transcriptional systems (Figs 5 and F, J-L in S1 Appendix). TwinCons can be used on any composite alignment and can identify similar segments between structurally similar or different proteins.

## Significance of rRNA and rProtein colocalized signatures for the deepest branching events in phylogenetic trees

TwinCons identifies signatures that determine the deepest and most ancient branching events in the tree of life (TOL). The ribosome is the ultimate molecular fossil, holding records of protein and RNA co-evolution [109–112]. Ribosomal RNA was the 'gold standard' for phylogenetic analysis for many years [113]. In the rRNA derived 'Woesian' tree Bacteria, Archaea, and Eukarya represent three major independent branches [11]. Recently, concatenated rProtein sequences have been used for determining and characterizing the TOL [8,114–117], suggesting that Eukarya may have branched from within the Asgard archaeal superphylum. Thus, phylogenetic trees built from rRNA and rProteins show substantially different topologies [8]. The deepest branches in the TOL are determined by signature positions in rRNA [92] and likely in rProteins.

TwinCons provides the toolkit to identify sequence signatures in both rRNA and rProteins. This analysis can be further substantiated if coupled with information derived from the 3D structure. Specifically, we identify the structural clusters, in which signatures in both rRNA

and rProteins (the deepest evolutionary signals) are colocalized as in the case of uL33 and its RNA binding site. Furthermore, it is possible to identify the asymmetric regions, in which the evolutionary changes in one polymer are not compensated by adjustments in the surrounding counterpart. We hope that the identification of such polarized regions combined with the structural and compositional analysis of the ribosomes will provide clues for evolutionary processes that resulted in divergence of archaeal and bacterial translation systems.

## Conclusion

TwinCons is a single metric that highlights conserved, variable, and diverging (signature) single column positions between a given pair of sequence groups within an alignment of protein or RNA. The TwinCons score can be used to query deep ancestry of short peptides from full protein alignments. Coupling TwinCons with supervised machine learning techniques provides a robust method for probing ancestry between any pair of possible protein candidates. TwinCons improves on existing signature-identifying methods by providing greater accessibility and flexibility in usage, in calculation methods, in visualization capabilities, and in simultaneous application for RNA and protein. TwinCons analysis coupled with information inferred from 3D structures can be a useful tool in studying colocalized changes in RNA and protein assemblies like the ribosome and uncovering mechanisms of sequence change and conservation hidden in ancient molecular history.

## Supporting information

**S1 Appendix. Supplementary text, figures, and tables. Fig A.** ROC curves for classifiers with different parameters built from the BaliBASE dataset, tested against the rProt dataset. Parameters shown here are segment boundaries (length threshold), TWC intensity for detection of positive positions (intensity threshold), and what percentage gaps should be used for removal of alignment columns (gap threshold). Each subplot represents different combination of the intensity and length thresholds. Colored lines within subplots represent different gap thresholds. Cutting only alignment positions with more than 80–90% gaps produce better distinction between true positive and true negatives. Complete data including all tested parameters and datasets is available in S2 Dataset. **Fig B**. ROC curves of training classifiers with different penalties and gamma parameters. The first four subplots (A-D) test different penalties and the last four test different gamma values (E-H). Each subplot represents a different dataset that was used for training and testing. (A) and (E) are PROSITE, (B) and (F) are BaliBASE, (C) and (G) are INDELible, (D) and (H) are rProtein dataset. For testing each dataset was split in 3 folds. Each fold produces an ROC curve, we plot the mean of the three results as single curve and plot the standard deviation of the true positive rate as a shaded region around it. Complete data is available in S3 Dataset. **Fig C**. ROC curves generated from HHalign alignments from the four datasets: BaliBASE, rProtein, INDELible, and PROSITE. Colored lines within subplots represent different gap thresholds used for column exclusion. **Fig D.** Comparison of structural mapping between Zebra2 and TwinCons. A) Zebra2 results and B) TwinCons results from sequence alignment for uL2 between archaeal and bacterial sequences mapped on the E. coli uL2 structure from PDB 4V9D [22]. C) Zebra2 results and D) TwinCons results from the same sequence alignment mapped on the P. furiosus uL2 structure from PDB 4V6U [36]. In panels A) and C) red indicates signatures. In panels B) and D) dark green indicates alignment positions with high conservation of residues, purple indicates signature positions, gray indicates heavily gapped regions in the composite alignment. Orange circles indicate signature positions. **Fig E.** TwinCons mapped for a short α-helix region in uL2 with analogous sequence between Bacteria and Archaea. Residues depicted here are listed in Table C in S1 Appendix.

(A) stick representation for E. coli uL2. (B) stick representation for P. furiosus uL2. (C) cartoon representation of E. coli uL2. (D) cartoon representation of P. furiosus uL2. (E) and (F) show different angle for the E. coli and P. furiosus uL2. Conserved residues are colored green, signatures are colored purple, and random positions are white. Heavily gapped regions, present in a single group, are colored gray. Figure generated with PyMOL. PDB IDs and chains used for the figure are available in Table E in S1 Appendix. **Fig F.** TwinCons segment with significant sequence similarity between (A, B) bL33 and (C, D) aL42. The segment is shown with full opacity cartoon, non-segment regions are shown with transparent cartoon. Conserved residues are colored green, signatures are colored purple, and random positions are white. Heavily gapped regions, present in a single group, are colored gray. Segment definitions are available in S6 Dataset. Figure generated with PyMOL. PDB IDs and chains used for the figure are available in Table E in S1 Appendix. **Fig G.** TwinCons score for Archaea and Bacteria composite alignments of the small and large subunits. (A) Secondary structure of the P. furiosus 16S rRNA with mapped TwinCons. (B) Secondary structure of the P. furiosus 5S and 23S rRNAs with mapped TwinCons. (C) Surface representation of the 16S rRNA for P. furiosus ribosome. (D) Surface representation of the 5S and 23S rRNAs for P. furiosus ribosome in crown view. Both the small and large subunits are shown from the subunit interface direction. Gray indicates heavily gapped regions, present only in bacterial or archaeal sequences; dark green indicates highly conserved regions between both bacterial and archaeal sequences; dark purple indicates signature regions between bacterial and archaeal sequences; white indicates sequence variable regions. In panels (A) and (B) blue numbers indicate helical numbering and ribosomal domains are indicated with brown. Panels (A) and (B) are generated with RiboVision, panels (C) and (D) are generated with PyMOL. PDB IDs and chains used for the figure are available in Table E in S1 Appendix. **Fig H.** TwinCons signatures differ based on the substitution matrix used. TwinCons results mapped on (A) metacaspase, (C) caspase, and (B) β-sheet superimposition of both structures, using the Blosum62 matrix. TwinCons results mapped on (D) metacaspase, (F) caspase, and (E) β-sheet superimposition of both structures, using structure-informed substitution matrices. A position with differing result is highlighted between panels (B) and (D) with red. Set of residues, representing the composite alignment column for the highlighted position, are shown between (B) and (E). Structure-informed matrices produce stronger signature signal between the two groups for this alignment position. Structures are generated with PyMOL. PDB IDs and chains used for the figure are available in Table E in S1 Appendix. **Fig I.** Distribution of TwinCons scores from the E. coli rRNA, based on three composite alignments between Archaeal and Bacterial sequences of 23S, 16S, and 5S rRNA. (A) Histogram of TwinCons scores showing three peaks of distribution around the minimum score, score zero, and the maximum score. (B) Scatter plot of TwinCons scores with group assignment by k-means clustering algorithm. The y-axis holds randomly assigned values and is only illustrative. Scores from different groups are colored with the viridis gradient. The red and green lines indicate the calculated thresholds of the groups spanning the lowest (red) and highest (green) scores. Thresholds calculated from each composite alignment are available in Table B in S1 Appendix. **Fig J.** TwinCons segments with significant sequence similarity between the P-loop domains of (A, C) aIF5 and (B, D) EF-Tu. Segments are shown with full opacity cartoon, while non-segment regions are shown with transparent cartoon. GDP from the EF-Tu structure is shown with sticks. Conserved residues are colored green, signatures are colored purple, and random positions are white. Heavily gapped regions, present in a single group, are colored gray. Segment definitions are available in S6 Dataset. Figure generated with PyMOL. PDB IDs and chains used for the figure are available in Table E in S1 Appendix. **Fig K.** TwinCons segment with significant sequence similarity between bS1 and domain 7 of RNAP mapped on the RNAP7 structure. (A) and (B) two views of the segment mapped on the

RNAP7 structure. Segment is shown with full opacity cartoon, while non-segment regions are shown with transparent cartoon. Conserved residues are colored green, signatures are colored purple, and random positions are white. Heavily gapped regions, present in a single group, are colored gray. Segment definitions are available in S6 Dataset. Figure generated with PyMOL. PDB IDs and chains used for the figure are available in Table E in S1 Appendix. **Fig L.** Twin-Cons segment with significant sequence similarity between bL34 and aL37. (A) representation of E. coli bL34, (B) representation of P. furiosus aL37, (C) 90-degree rotation view of E. coli bL34, and (D) 90-degree rotation view of P. furiosus aL37. The segment is shown with full opacity cartoon, non-segment regions are shown with transparent cartoon. Conserved residues are colored green, signatures are colored purple, and random positions are white. Heavily gapped regions, present in a single group, are colored gray. Segment definitions are available in S6 Dataset. Figure generated with PyMOL. PDB IDs and chains used for the figure are available in Table E in S1 Appendix. **Table A.** Substitution matrices available for TwinCons calculation and references for full descriptions. **Table B.** TwinCons thresholds calculated with 5 k-clusters for different subsets of rRNA. First two rows, tagged with 'ribosome', include sequences from the 23S, 5S, and 16S. Entries tagged with LSU include sequences from the 23S and 5S. Entries tagged with SSU include only rRNA from the 16S rRNA. TwinCons was calculated against the Archaea-Bacteria composite alignment of the rRNA. Standard deviations were calculated after repeating the calculation 100 times. Full script used to generate this data can be found at https://github.com/LDWLab/TWC_distribution. **Table C.** TwinCons and ConSurf statistics for α-helical region in uL2. Positions with low Shannon entropy, low ConSurf score, and high TwinCons score are detected as highly conserved. Positions with TwinCons below -0.6 are detected as signature positions. Signature positions detected with TwinCons, that are detected as conserved by ConSurf are highlighted with blue. **Table D.** Composite alignments used in sequence similarity analysis. **Table E.** Protein and rRNA structures used to map sequence similarity analysis. When multiple PDBs are used in a single row they are separated by a semicolon. When multiple chains are used from a single PDB they are separated by &.
(PDF)

**S1 Dataset. Table with BaliBASE alignment names with enzyme and EC annotations present in the alignment.** Alignments with similar EC annotations are colored the same. Combinations between alignments that share color are excluded from dataset generation.
(XLSX)

**S2 Dataset. Figures with ROC curves for all tested parameters.** The title of each figure indicates the training dataset and the tested dataset. For example, "BBS vs PROSITE" indicates that the training set was BaliBASE and it was tested against the PROSITE dataset. ROC labels of subplots and lines are the same as the ones used in Fig A in S1 Appendix.
(PDF)

**S3 Dataset. Data used to generate Fig B in S1 Appendix.** The title of each sheet indicates whether penalties or gamma values were tested. TPR, FPR, and TPR standard deviation for each of the datasets. Calculations were done for boundary distance thresholds varying from -20 to 20 with a step of 0.05.
(XLSX)

**S4 Dataset. Performance of trained classifier from the BaliBASE dataset with best parameter combination against itself and the three other datasets.** TPR, TNR, and precision for boundary distance thresholds varying from -5 to 5 with a step of 0.1. The distance thresholds of 0.7 and 1.5 are highlighted.
(XLSX)

**S5 Dataset. Description of query composite alignments used in Fig 4.** Alignments are available at https://apollo2.chemistry.gatech.edu/TwinConsDatasets/.
(XLSX)

**S6 Dataset. TwinCons segment results for composite alignments used in Fig 4.** Each segment is identified with its alignment position and the distance it was from the decision boundary.
(CSV)

**S7 Dataset. TwinCons results from rRNA composite alignment of 23S, 16S, and 5S sequences between Archaea and Bacteria.** TwinCons results from protein composite alignments of caspase-metacaspase and uL2. Thresholds for signature and conserved positions for each alignment are indicated.
(XLSX)

**S8 Dataset. INDELible control file used to generate artificial sequence alignments from random sequence seeds, evolved under biological model.**
(TXT)

**S9 Dataset. Combined TwinCons and HHalign results for segments detected within the query alignment set.** The file reports alignment group names, TwinCons scores and probability, TwinCons segment ranges, HHalign scores and probabilities, HHalign ranges, and index sequences used for the ranges.
(XLSX)

**S10 Dataset. Direct score comparison for the alignment of uL2 between TwinCons and Zebra2, as well as TwinCons and ConSurf.**
(XLSX)

## Acknowledgments

We thank prof. Jordan I. King and Dr. Gene Lamm for helpful discussions. We also thank Dr. Stephen Altschul and Dr. Yi-Kuo Yu for providing the matrix compositional adjustment code.

## Author Contributions

**Conceptualization:** Petar I. Penev, Anton S. Petrov.

**Data curation:** Anton S. Petrov.

**Formal analysis:** Petar I. Penev, Claudia Alvarez-Carreño, Anton S. Petrov.

**Funding acquisition:** Anton S. Petrov, Loren Dean Williams.

**Methodology:** Petar I. Penev, Claudia Alvarez-Carreño, Eric Smith, Anton S. Petrov.

**Project administration:** Anton S. Petrov.

**Software:** Petar I. Penev.

**Supervision:** Anton S. Petrov, Loren Dean Williams.

**Validation:** Petar I. Penev, Claudia Alvarez-Carreño.

**Writing – original draft:** Petar I. Penev, Anton S. Petrov, Loren Dean Williams.

**Writing – review & editing:** Petar I. Penev, Eric Smith, Anton S. Petrov, Loren Dean Williams.

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
