## [Decision Letter · Decision Letter 0]

1 Jul 2021

Dear Dr Petrov,

Thank you very much for submitting your manuscript "TwinCons: conservation score for uncovering deep sequence similarity and divergence" for consideration at PLOS Computational Biology.

As with all papers reviewed by the journal, your manuscript was reviewed by members of the editorial board and by several independent reviewers. In light of the reviews (below this email), we would like to invite the resubmission of a significantly-revised version that takes into account the reviewers' comments.

We cannot make any decision about publication until we have seen the revised manuscript and your response to the reviewers' comments. Your revised manuscript is also likely to be sent to reviewers for further evaluation.

Sincerely,

Christos A. Ouzounis

Associate Editor

PLOS Computational Biology

Arne Elofsson

Deputy Editor

PLOS Computational Biology

Reviewer's Responses to Questions

**Comments to the Authors:**

Reviewer #1: Manuscript: TwinCons: conservation score for uncovering deep sequence similarity and divergence

The manuscript presents TwinCons, a scoring function for evaluating the match between two amino acid (nucleotide) distributions observed at each position of two corresponding aligned sequence families (groups). Highly positive scores indicate the matching conservation across the families. Highly negative scores, called signatures in the manuscript, show conservation high within the families but divergent across them. While high conservation across the families indicates evolutionarily important sites, signatures may specify functionally important sites for one of the families. The TwinCons usefulness lies in the possibility to apply it to protein and nucleic acid multiple sequence alignments (MSAs). The manuscript presents results from the application of TwinCons to both types of MSAs.

Comments:

1. TwinCons depends on a substitution matrix (SM) M in Eq. 2. Can TwinCons use multiple SMs? If so, how does it combine them?

2. The section "Generating random alignments" (line 273) lacks details of how random alignments were generated: how randomized sequence seeds were obtained and how a random tree was constructed, leading to a random alignment.

3. The manuscript should provide information on how the MSAs were obtained for all examples in the results section.

4. It would be useful to see both structures for the full context of the correlation between sequence-based scoring and structural observations in the section "TwinCons detects highly conserved and signature positions in composite alignments" (line 429).

Also, as the manuscript benchmarks TwinCons against these methods for other use cases, a comparison with Zebra2 and HHalign conservation and signature scores (highly negative scores for HHalign), and similarly, a comparison with ConSurf and HHalign in the section "TwinCons provides convenient representation of signatures and conservation" (line 463), would be useful.

5. Table 1 in the section "TwinCons: score that detects sequence similarity between pairs of proteins" (line 500) provides a comparison with HHalign for 8 out of 36 alignments (line 511). Would it be possible to include all alignments in the comparison?

6. Would it be possible to evaluate the accuracy of the tested methods, e.g., by quantifying the number of true and false positives and negatives for identified signatures?

7. The section "Comparison of TwinCons signature positions with Zebra2" states that a "structure-informed substitution matrix" was used for one of the use cases (line 734). The optimal configuration, however, includes the BLOSUM62 matrix (line 346). Was the same SM used in the application throughout the results section? If not, what criteria were used for selecting a substitution matrix for application?

Reviewer #2: TwinCons is a score for evaluating conservation patterns within and between two groups of sequences. The authors present numerous examples of the methods utility for distinguishing conserved, nonconserved, and "signature" residues. The ability to visualize conserved and unique resides on a PDB structure is particularly nice.

## Major Critiques

1. Substitution matrices are typically in logorithmic units. Why is a linear multiplication with the frequency vector appropriate? More broadly, I think additional theory tying the TWC score to the statistics of substition processes is required. For instance, if the matrix M were a Markov chain transition matrix (such as PAM scores, when properly converted to linear space), then the quantity Mx represents the expected drift of the sequence over a fixed period of time. The product yMx is then the dot product of the two frequency vectors after accounting for a fixed period of drift. Of course, for substitution matrices which are not transition matrices this theory is not valid, but in that case I think a deeper discussion could motivate the choice of scoring matrix.

2. Uniform distributions are used for the null model in gaps and in the matrix scaling. However, uniform distributions are not very realistic for biological sequences. Would it be better to use background frequencies instead? In some cases these can be derived from the substitution matrix (e.g. the eigenvectors of a transition matrix give the stationary distribution).

3. More details are needed about the matrix scaling procedure. "Uniform scaling" (line 207) would normally suggest multiplication by a scalar factor, but to produce the expected 0 value I suppose an additive component must be used. Is there any additional scaling done between matrices to assure that the range of TWC values are comperable, in addition to having a consistent zero point? This issue is particularly important when comparing RNA and protein TwinCons scores, and may account for the saturation of rRNA scores.

4. The segment creation rule of two negative scores seems arbitrary. Were any other rules considered? The zero point for a particular set of parameters is going to be an approximation, both due to the scaling issue in point 3 and to the generous rounding used in many substitution matrices. It seems like a rule which included the magnitude as well as the sign would be more robust and might decrease fragmentation (lin 523).

5. Was SVM with rbf kernel the only classifier considered? Since the data is highly non-seperable SVM does not seem an obvious choice. Distance from the boundary is also not a very good metric for the quality of a segment. A probabilistic classifier would give a better calibrated quality score.

6. The procedure for rRNA segment creation appears to be significantly different from the protein method. The Suppl Text is a bit unclear and should be revised, but I think the fact that different thresholds were used for RNA should be discussed in the main text. The suppl text refers to k-mers in the context of k-means clustering; probably "k clusters" was intended (also S2).

7. One major limitation of TwinCons is that it relies entirely on the input alignment and the division into two groups. This is discussed towwards the end (735), but I think it should be emphasized more. In particular, the source of the alignment for all examples should be clearly stated. The non-permuted bL33/aL42 alignment should be pointed out to be deliberately wrong (e.g. to align non-homologous residues), as it is presumably a low-scoring artifact.

## Minor Critiques

8. Several sentences could be naively interpretted that TwinCons allows one to compare two nonhomologous genes/proteins (e.g. 'inspecting a pair of polymers' (35), 'correlated changes in RNA and protein assemblies' (778), etc). Particularly in the abstract it should be clear that TwinCons applies to alignment rows and does not address coevolution.

9. I suggest using a diverging color palette for decision boundary distance (e.g. Fig 1C) to emphasize the boundary. (https://seaborn.pydata.org/tutorial/color_palettes.html#perceptually-uniform-divering-palettes)

10. The gradients in Fig 1B make it difficult to read the bar plot.

11. The 'TwinCons Score' Section (line 115) gets into some details that are not explained until the methods section. Please consider the organization.

12. "random residue distribution" should be "uniform" (line 208) to prevent confusion with background frequencies.

13. Coefficients & formula for the trained SVM should be included (may be supplemental).

14. A table comparing the different data sets used for SVM training might be helpful (number of positive & negative examples, whether negatives were filtered, etc).

15. Why are some sidechains visible in Fig 3? If they are important residues, perhaps they should be shown consistently across all structures.

16. If the goal of Fig 2 & 3 is to compare TwinCons to ConSurf and Zebra2 respectively, I wonder if a dotplot showing the correlation between scores directly would be more pertinent. This could be quite interesting, as comparing the different color scales can be difficult by eye.

17. The application to rProtein and rRNA trees (lin 677) was not clear to me. How can TwinCons be applied to more than one gene at a time? Particularly cases with different alphabets?

18. I don't understand the analogy to dipole energy (line 706). Is this just the fact that both are vector quantities?

## Code

19. I was able to successfully install and run the tool. Code is sufficiently documented and available in the pypi repo. The URL needs to be updated to https://github.com/LDWLab/AlignmentScore. It might make sense to rename the project to LDWLab/TwinCons or similar to make it easier to find.

20. It would be nice to include in the repository a complete example. For instance, there are no pdb files available for use with TwinCons.py

**Have the authors made all data and (if applicable) computational code underlying the findings in their manuscript fully available?**

Reviewer #1: Yes

Reviewer #2: Yes

PLOS authors have the option to publish the peer review history of their article (what does this mean?). If published, this will include your full peer review and any attached files.

Reviewer #1: **Yes: **Mindaugas Margelevicius

Reviewer #2: **Yes: **Spencer Bliven
---

## [Decision Letter · Decision Letter 1]

6 Oct 2021

Dear Dr Petrov,

We are pleased to inform you that your manuscript 'TwinCons: conservation score for uncovering deep sequence similarity and divergence' has been provisionally accepted for publication in PLOS Computational Biology.

Best regards,

Christos A. Ouzounis

Associate Editor

PLOS Computational Biology

Arne Elofsson

Deputy Editor

PLOS Computational Biology

Reviewer's Responses to Questions

**Comments to the Authors:**

Reviewer #1: The authors have addressed my comments.

Reviewer #2: The authors have made substantial improvements to the method, analysis, and the text. On every point from my previous review the authors have addressed the issues. I particularly appreciate the detailed theory that has been added in the supplemental.

## Minor Issues

- There's a duplicated sentence in the suppl text (pg 2, These differences of...)

- In point 16 from my previous review I mistakenly suggested a "dotplot" when I meant a simple scatter plot comparing TwinCons vs the other scores. This would show the correlation between scores (e.g. how low ConSurf scores correlate to both positive and negative TwinCons). However, with the addition of the ul2 data in S10 it is easy for interested readers to do this for themselves.

**Have the authors made all data and (if applicable) computational code underlying the findings in their manuscript fully available?**

Reviewer #1: Yes

Reviewer #2: Yes

PLOS authors have the option to publish the peer review history of their article (what does this mean?). If published, this will include your full peer review and any attached files.

Reviewer #1: **Yes: **Mindaugas Margelevičius

Reviewer #2: **Yes: **Spencer Bliven

---

## [Editor Report · Acceptance letter]

22 Oct 2021

PCOMPBIOL-D-21-00651R1 

TwinCons: conservation score for uncovering deep sequence similarity and divergence

Dear Dr Petrov,

I am pleased to inform you that your manuscript has been formally accepted for publication in PLOS Computational Biology. Your manuscript is now with our production department and you will be notified of the publication date in due course.

With kind regards,

Katalin Szabo
